# Predicting the Digestive Tract Development and Growth Performance of Goat Kids Using Sigmoidal Models

**DOI:** 10.3390/ani11030757

**Published:** 2021-03-10

**Authors:** Mahmoud Abdelsattar, Yimin Zhuang, Kai Cui, Yanliang Bi, Naifeng Zhang

**Affiliations:** 1Key Laboratory of Feed Biotechnology of the Ministry of Agriculture, Feed Research Institute, Chinese Academy of Agricultural Sciences, Beijing 100081, China; m.m.abdelsattar@agr.svu.edu.eg (M.A.); zym1164323345@163.com (Y.Z.); cuikai@caas.cn (K.C.); vetbi2008@163.com (Y.B.); 2Department of Animal and Poultry Production, Faculty of Agriculture, South Valley University, 83523 Qena, Egypt

**Keywords:** goat kids, rumen development, growth curve, inflection point, area under the curve

## Abstract

**Simple Summary:**

Growth is a biological phenomenon that usually happens with increased mass or volume over time. The transition from birth to postweaning has a fundamental impact on characteristics specific to the development and growth of the digestive tract. Consequently, the growth performance in goat kids is due to the combined effects of age, rumen development, and feeding system, which controls the growth patterns of each stage. Generally, the development of rumen is in stages. Therefore, modeling informative growth curves is of great importance for a better understanding of effective development patterns for optimizing feeding management systems for improved production efficiency. For the above reasons, this study was conducted to compare the sigmoid and polynomial models which were used to determine the age-related changes in body weight, body size indices, and digestive tract development; and to identify the most appropriate model. These empirical mathematical models of growth patterns are continuous functions of time and have biologically meaningful parameters that can be used to predict growth efficiently. In light of these models which lack sufficient information in literature, further research about different species of goats in different environments is still needed, thereby enabling us to distinguish more accurate results.

**Abstract:**

The transition from monogastric to rumination stage is crucial in ruminants’ growth to avoid stressors—weaning and neonatal mortalities. Poor growth of the digestive tract could adversely affect the performance of the animal. Modeling informative growth curves is of great importance for a better understanding of the effective development pattern, in order to optimize feeding management system, and to achieve more production efficiency. However, little is known about the digestive tract growth curves. For this reason, one big goat farm of Laiwu black breed was chosen as a basis of this study. Forty-eight kids belonging to eight-time points (1, 7, 14, 28, 42, 56, 70, and 84 d; 6 kids for each) were selected and slaughtered. The body weight, body size indices, rumen pH, and stomach parts were determined and fitted to the polynomial and sigmoidal models. In terms of goodness of fit criteria, the Gompertz model was the best model for body weight, body oblique length, tube, and rumen weight. Moreover, the Logistic model was the best model for carcass weight, body height, and chest circumference. In addition, the Quadratic model showed the best fit for dressing percentage, omasum weight, abomasum weight, and rumen volume. Moreover, the cubic model best fitted the ruminal pH and reticulum percentage. The Weibull model was the best model for the reticulum weight and omasum percentage, while the MMF model was the best model describing the growth of chest depth, rumen percentage, and abomasum percentage. The model parameters, R squared, inflection points, area under curve varied among the different dependent variables. The Pearson correlation showed that the digestive tract development was more correlated with age than body weight, but the other variables were more correlated with body weight than age. The study demonstrated the use of empirical sigmoidal and polynomial models to predict growth rates of the digestive tract at relevant age efficiently.

## 1. Introduction

China with 18.19% of the total world goat population, has superiority in the goat production sector, which is increasing rapidly [1]. China is a leader country in goat meat and skin production, producing about 35.89% and 30.41% of the total world production, respectively [1]. The milk production of dairy goats in China is 2% of the world’s goat milk production [2]. The Laiwu black goat used in the present study is indigenous to Shandong, nearly 230,000 heads and mainly reared for cashmere, pelts, and meat [3].

Increasing body weight and feed efficiency are of importance for profitable goat farming [4]. The growth of body weight and body weight measurements depends on the digestive tract’s development [5,6]. In goats, the gastrointestinal tract as a percentage of the empty body weight increased from 13 up to 27% as they grew, in which the rumen contributed most to the growth [6]. As the ruminant matures, the rumen as a percentage of the whole forestomach increases from 40 to 80%, while the other digestive organs decline [7]. The development of rumen is a substantial physiological challenge for young ruminants, which has significant impacts on their entire life [8,9]. The transition period to gain full rumen functionality was long from 16.9 up to 25.9 kg body weight in goats [6]. Complete functional development of rumen is a prerequisite for weaning and is an indicator of digestion efficiency and stability [10,11].

After birth, the rumen has undeveloped volume and function, and the esophageal groove allows milk to enter the abomasum and keeps rumen out of milk [12]. The milk intake gradually decreased and the solid intake increased due to progressive independent behavior from the dam over time [13]. Solid feed had significant impact on the rumen development via increasing the rumen volatile fatty acids [8,14]

The livestock growth curve has a sigmoid shape so that it can be fitted and analyzed by the nonlinear mathematical model. The use of mathematical growth models provides a good way of summarizing the growth potential’s information into a small set of parameters [15]. These estimated parameters are biologically meaningful and can be used to envisage growth patterns over time and to predict the growth of one variable by knowing the other variable [16]. There are usually two types of growth functions used to describe evolution over time. The first is a sigmoid curve with a fixed inflection point (e.g., logistic and Gompertz) [17,18]. The logistic inflection point is symmetric at half of the theoretical maximum body weight. In contrast, the Gompertz inflection point is asymmetric and calculated as the theoretical maximum body weight divided by the e value [19]. The second is a sigmoid curve with a flexible asymmetric inflection point (e.g., Morgan-Mercer-Flodin –MMF-, Richards, and Weibull) [20,21]. The MMF is derived from consideration of thermodynamic changes, while Weibull is not derived from thermodynamic functions, but it also has good applicability.

The polynomial regression models (Quadratic and Cubic) can also be used for the prediction of the growth rates at relevant time points [22]. No assumptions regarding the shape of the curve are made in the polynomial models which allows more flexible growth curves than some of the nonlinear models [23].

Several studies used many mathematical functions to represent growth performance over time in goats [6,24,25,26,27]. For example, Campos et al. [6] studied the prediction of empty body weight (the weight of goats minus the digestive tract contents). In avian, the mathematical models were used to predict the growth rate and size of the alimentary tract [28]. The authors found a positive relationship between the growth rate and the size of the alimentary tract. There are conflicting reports on the models for describing the digestive tract’s growth patterns in goat kids. Fitting curves that resemble the observed growth of stomach parts will be useful for further feed regime modification, breeding practices, and better selection of weaning age. Therefore, the objectives of the current study were; to model growth curves considering sigmoidal, and polynomial equations; and to compare the accuracy of fitness between the models to recognize the best model. We hypothesized this study would cover the shortfall of information about the growth curves of digestive tract anatomic development in the goat kids.

## 2. Materials and Methods

### 2.1. Animal Trial and Feeding Management

The experiment was carried out on 9 November 2019 at Xiang Feng black goat farm, Jinan, China. A total of forty-eight healthy female Laiwu black goat kids were randomly selected from specific age groups (1, 7, 14, 28, 42, 56, 70, and 84 d; each age had as 6 replicate). As shown in Figure 1, the kids lived with dams and consumed colostrum from birth to d 7 then milk from d 8 to d 28. After 28 d of age, the concentrate was introduced to the kids as their supplementary diet. The kids were weaned at 60 d of age and separated from their dams. Accordingly, the solid feed was the only food source for the goats after weaning. Solid diets were formulated to meet their nutrient requirements of goat kids [29]. Kids were fed the solid twice daily at 8:00 and 16:00, respectively. The ingredients and the chemical compositions of the solid diet were listed in Table 1. In addition, the goat kids were raised in standard pens with a slatted floor, with up to 20 animals per pen. Kids lived with their dams until 60 d of age, and then the kids were weaned and moved to growing pens. Feed and water were changed daily and provided *ad libitum*.

### 2.2. Anatomic Development Estimations

The selected forty-eight goat kids were weighted before feeding. Body size indices of goat kids, including body height, oblique length, tube, chest circumference, and chest depth, were measured using a tape. Afterwards, the goat kids were slaughtered at the farm slaughterhouse, according to the Chinese industrial practices. The dressing was performed by removing the external and internal organs. The hot carcass weight was weighed using a scale within an accuracy of 50 g. The dressing percentage was calculated as the percentage of hot carcass weight as a proportion of live body weight. Furthermore, the stomach parts (rumen, abomasum, omasum, and reticulum) were dissected and weighted after the digesta was removed. These digestive parts were expressed as a percentage of the complex stomach weight. In addition, the rumen volume was measured by filling the rumen with water. Moreover, the pH was measured in the rumen fluid immediately after opening the abdominal sac using a pH automatic detector.

### 2.3. Statistical Analysis

A selection of 6 sigmoidal models included three-parameter models (Logistic, Gompertz, and Ratkowsky) and four-parameter models (MMF, Weibull, and Richards), as well as two polynomial models (Quadratic and cubic), were fitted to the measurements of goat kids growth related with age. The expressions of models are shown in Table 2. All models were fitted using the Curve Expert Professional Program (ver. 2.6.5; Levenberg-Marquardt method). Within these growth equations, the observed data of growth (y) were fitted as a function of the independent variable (x, age). All models were assessed to recognize the favored model, which can meticulously describe the data. The factors used to compare the models including the mean square error (SEM), Akaike’s information criteria (AIC) [30], corrected Akaike’s information criteria (AICc) [31], and Bayesian information criterion (BIC) [32], according to the following Equations:(1)$$\mathrm{MSE}=\frac{\sum_{i=1}^{n}{\left(y_{i}-{ŷ}_{i}\right)}^{2}}{n-2}$$
(2)$$\mathrm{AIC}=n\times \ln(\frac{\mathrm{SSR}}{n})+2k$$
(3)$$\mathrm{AICc}=n\times \ln(\frac{\mathrm{SSR}}{n})+2k\times \frac{n}{n-k-1}$$
(4)$$\;\mathrm{BIC}=n\times \ln(\frac{\mathrm{SSR}}{n})+k\ln\left(n\right)$$
where $y_{i}$ is ith observed values,$\;{ŷ}_{i}$ is ith estimated by the model, n is the number of observations, k is the number of estimated parameters, and SSR is the sum of squares of residuals of the model. The calculation of AIC, AICc, and BIC assumed that the model errors were independent with normal distribution.

The model that showed the best goodness of fit is commonly chosen based on mean square error AIC, and BIC [33,34]. However, AICc could also estimate the optimum model, although it is less applicable than AIC because its justifications depend upon the candidate model’s shape, such as the number of parameters. Generally, the lower values of these estimations reflect the best fitting.

Then, the model that showed the best goodness of fit was selected to estimate the model parameters, coefficient of determination (R^2^), weight at the inflection point (IPy), and age at the inflection point (IPx). In addition, the model could predict the area under the curve (AUC) between different critical points. AUC could describe the variation of growth as a function of time. AUC as a biological value is an important derivative of the models which was calculated as follows:(5)$$\mathrm{AUC}=\overset{b}{\underset{a}{\int }}y\left(x\right)\;\mathrm{dx}$$
where y(x) is the curve given function having the limits of x = a and x = b,

In addition, the Pearson correlations between the different variables were performed using SPSS software (SPSS Inc., Chicago, IL, USA).

## 3. Results

### 3.1. Performance and Carcass Measurements

The goodness of fit of the growth curve models for body weight, carcass weight, dressing percentage, and body size indices is shown in Table 3. The selected models are shown in Table 4 and Figure 2. The MSE, AIC, AICc, and BIC criteria were used to discriminate among the equations for the best one. The Gompertz model was the best model for body weight as it had the lowest AICc, AIC, BIC compared to other models (AICc = 19.88, AIC = 19.60, and BIC = 23.30). The inflection point were at 6.85 kg and 45.93 d. The Quadratic model for the dressing percentage showed the lowest MSE (3.39), AICc (115.97), AIC (115.70), and BIC (119.40). The Logistic and Ratkowsky were the best models for carcass weight (AICc = −21.41, AIC = −21.68, and BIC = −17.98), body height (AICc = 111.79, AIC = 111.51, and BIC = 115.21), and chest circumference (AICc = 97.29, AIC = 97.02, and BIC = 100.72). The weight and age at the inflection point were 2.72 kg and 27.35 d for carcass weight, 27.29 cm and -17.90 d for body height, and 32.14 cm and −4.08 d for chest circumference, respectively. The Gompertz model was the best model for of oblique length (AICc = 131.38, AIC = 131.11, and BIC = 134.81), and tube (AICc = −43.80, AIC = −44.07, and BIC = −40.37). The growth curve parameter IPy and IPx were found to be 19.60 cm and −18.14 d for oblique length, 2.80 cm, and −59.51 d for the tube, respectively. In addition, the MMF model was the best fit for the chest depth (MSE = 3.05 and AIC = 106.68). The chest depth reached its inflection point at 12.36 cm and 19.56 d.

The AUC of body weight, carcass weight, and body size indices were increased over time. However, the AUC of dressing percentage did not show differences over time. In addition, the R^2^ values for body weight, carcass weight, oblique length, height, and both chest depth and circumference were equal or higher than 0.69. While the R^2^ values for the dressing percentage and tube were equal or lower than 0.32. Moreover, the Pearson correlation was positive between the above mentioned variables and both of age and body weight. However, the dressing percentage did not show significant correlation with age or body weight (Table 5).

### 3.2. Anatomic Development of Digestive Tract

The goodness of fit of sigmoid and polynomial equations which were used to model the growth characteristics of the stomach compartments, rumen volume, and pH is shown in Table 6. The model which considered as the best choice for describing the development of kids’ complex stomach proportions is shown in Figure 3 and Figure 4. In addition, the estimated parameters, inflection points, and AUC of the selected models are shown in Table 7. The overall statistics showed that the Cubic model best described the data of the rumen pH (MSE = 0.52, AICc = −58.52, AIC = −59.08, and BIC = −53.53) and the reticulum percentage (MSE = 2.57, AICc = 91.09, AIC = 90.53, and BIC = 96.08).

In addition, the Quadratic model was determined to be the best model describing the evolution of the rumen volume (MSE = 148.67, AICc = 471.34, AIC = 471.06, and BIC = 474.76), the omasum weight (AICc = 131.26, AIC = 130.99, and BIC = 134.69), and the abomasum weight (AICc = 209.97, AIC = 209.69, and BIC = 213.39). In addition, the Gompertz model was the best model for the rumen weight (SEM = 17.35, AICc = 269.43, AIC= 269.16, and BIC = 272.86). The inflection point for rumen weight was at 601.10 g and 157.46 d. Moreover, the Weibull model showed the best fitting for the reticulum weight (SEM = 4.76, AICc = 149.00, AIC = 148.44, and BIC = 153.99) and the omasum percentage (SEM = 1.96, AICc = 65.69, AIC = 65.14, and BIC = 70.69). The inflection point for the reticulum weight was at 17.54 g and 45.61 d. In addition, the inflection point for the omasum percentage was at 7.97% and 61.78 d. Furthermore, the MMF model was the best model for rumen percentage (SEM = 6.64, AICc = 180.34, AIC = 179.79, and BIC = 185.34) and abomasum percentage (SEM = 6.43, AICc = 177.32, AIC = 176.76, and BIC = 182.31). The inflection points occurred at 33.38% and 18.73 d for rumen and at 47.42% and 23.78 d age for abomasum.

The AUC for the stomach parts increased over time. However, the abomasum percentages had higher AUC before weaning than after weaning. Moreover, the AUC of pH was higher from birth to 28 d than the other stages. In addition, the R^2^ values for rumen volume, rumen percentage, and abomasum percentages were equal or higher than 0.63. While the R^2^ values of pH, reticulum percentage, and omasum percentage were 0.15, 0.31, and 0.46, respectively.

Fundamental changes have occurred in the rumen and abomasum percentages over time (Figure 4). The rumen percentage was increasing and the abomasum percentage was decreasing over time. The development of the rumen and abomasum could be divided into three stages. In the first stage (birth), the abomasum tended to decrease, but the rumen tended to increase. In the second stage, the curve lines crossed at the intersection point. At the final stage, the rumen increased constantly, but the abomasum decreased rapidly. The slope of the curve indicated that the growth changes were highly discernible during the period from 14 d to 56 d of age. In Table 8, the age and body weight were positively correlated with the digestive tract development instead of the abomasum percentage which had a negative correlation with both age and body weight (−0.90 ** and −0.81 **, respectively).

## 4. Discussion

### 4.1. Body Weight, Carcass Weight, and Dressing Percentage

The study objective was to compare the goodness of fit of both sigmoidal and polynomial functions to provide a specific shape of the growth curve from birth to 84 d of age in goat kids. The interpretation of growth is wide-ranging according to the type of model being used and the obtained data which differ according to the breed genotype and the environmental variations. The models are typically used to describe the growth, although each model has different characteristics and limitations [16]. Choosing a poor-fitting model produces unrealistic growth curves with futile growth rates, inflection points, and parameters [34].

In order to confirm the model validity, AIC is one of the crucial indicators for the goodness of fit, and the complexity of the model structure should be also considered [35]. In the current study, the Gompertz and Logistic models produced an excellent fit for different variables in goat kids than the linear models. It has been indicated that the non-linear models are more effective than linear models, because of their sigmoid shape [36]. The Gompertz model is an appropriate model describing the growth in local Tunisian goat kids [37], Beetal goats [25], Raeini Cashmere goat [38], Alpine goat [39,40], local Tunisian goat, and Damascus goats [39]. Other studies in lambs confirmed the Gompertz model provided the best fit for describing the growth data [22,41]. The logistic model showed a slightly best fitting for the growth of Repartida goats than Gompertz [42]. Moreover, Malhado, et al. [43] showed that both Gompertz and Logistic models provided the best fit of the growth curve in sheep. In addition, the logistic and Gompertz showed the best fit for the growth of young bulls with fewer iterations needed to achieve convergence [44].

These sigmoidal growth models were used extensively to describe the growth of different animal species, and summarize the growth measurements into few variables. Parameter a is the asymptotic weight, which refers to the maximum growth change according to the environmental and genetic effects. In the current study, the asymptotic weight was 18.61 of fitting body weight over time using the Gompertz model. The asymptotic weight was 17.97 in Raeini Cashmere goat [38] and 23.39 in Beetal goat [25]. The parameter b is a scale parameter which does not have biological interpretation. The parameter b value of body weight was 0.62, which was lower than the obtained value by fitting the Gompertz equation in Raeini Cashmere goat (1.97) [38] or Beetal kids (1.98) [25]. The parameter c is a maturing index that represents the maturity rate and the speed of growth in which the initial weight reaches the asymptotic weight- the higher the values of parameter c the earlier the maturity [38]. In this study, the c value of body weight was 0.01 and same as the value in Raeini Cashmere goat [38] but lower than that obtained for Beetal kids [25]. The parameter d is gives the shape of growth curve and determines the position of the curve point inflection. Both logistic and Gompertz models had three parameters than the four-parameter sigmoidal models such as Weibull, Richards, and MMF.

Point of inflection at which the concavity changes from up to down is useful for the evaluations of feed requirements, management systems, and breeding processes [45]. Some studies emphasized that the inflection point could be used to determine the onset of puberty [46], while others showed that it could determine the optimum slaughter age [47]. In the present study, the inflection point for body weight was at 45.93 d and 6.85 kg. In another study, the age and weight at the inflection point for Raeini Cashmere goat were 52.94 d and 6.63 kg, respectively [38]. However, the points of inflection in the current study and in Raeini Cashmere goat do not reflect the actual beginning of the auto-deceleration stage of growth.

The AUC of body weight, carcass weight, and body size indices increased over time, reflecting the physiological and morphological development of organs responsible for the growth over time, such as the digestive tract.

The Pearson correlation was positive between the body weight and carcass weight. Similar results showed that the slaughter age or weight had a significant relationship with carcass weight in goat kids [34]. Furthermore, there was a remarkable positive correlation between live body weight and hot carcass weight in Angora goats slaughtered at six years old [48]. In addition, the carcass weight and the empty body weight were higher in 10 kg kids compared to those of 6 kg kids [49]. In addition, the body size indices had positive correlation with the slaughter age and weight. The slaughter age progression boosted the carcass length, hind leg width, and shoulder in boar kids [50].

However, the Pearson correlation was not significant between the dressing percentage and animal weight or age. Supporting these results, several authors showed no differences in dressing percentages of kid goats slaughtered at different live weight [50,51,52,53]. However, other studies showed a positive relationship between dressing percentage and age in goat kids [34,54] and lambs [55].

### 4.2. Anatomic Development of Digestive Tract

The growth of the digestive tract and the body weight in the newborn ruminants has deterministic effects on the mortality rate and the production enterprise’s success. To the best of our knowledge, this is the first study to model the development of digestive tract in goats. In general, the Weibull function model produced an excellent fit for the reticulum weight and the omasum percentage based on the lowest information criteria value. The sigmoid functions with more parameters allow modeling more flexible S-shaped curves. In another study, fitting random sigmoidal models for the growth in Akkeci female white goats showed that the Weibull growth model was the best model for providing the most suitable values [56]. The cubic model was selected as the best model for pH. The non-linear models had worse performance than polynomial models in minipig [57]. The linear, quadratic, cubic, and exponential models were the best models to describe the mammary gland’s growth in goats [58]. At the same time, the logistic curve was the best sigmoidal model to describe the pH curve. Similarly, the logistic model was considered the optimal fit to describe pH curves in cattle [59,60]. The goodness-of-fit for growth of rumen and abomasum percentages showed that the best fit was reached using the MMF model. In Mengali sheep, MMF was the best-fitted model for the growth curve [36].

The asymptotic values of rumen and abomasum percentages obtained by the MMF model were 22.52 and 60.93, respectively. The higher asymptotic value of abomasum percentage indicates that it was heavy as adults and grew slowly. Thus, the maturing indexes of the rumen and abomasum percentages were 61.55 and 17.99, respectively. However, a and c parameters for the weight of rumen were higher than the abomasum due to the low initial abomasum weight.

The inflection points of the MMF model for rumen percentage occurred at 33.38% and 18.73 d, and for the abomasum percentage occurred at 47.42% and 24 d. We hypothesized that the best time for weaning could be inferred from the inflection point. The goat kids have been successfully weaned at 10 kg at 30 d of age [61]. In another study, goat kids can be weaned as early as four weeks of age [62]. However, the abrupt weaning at 10 kg caused a decrease in growth rate of *ad libitum* milk feeding goat kids [63]. In contrast, the appearance of stress depends on the management system and husbandry practices of the farm. The sudden weaning did not show adverse effect in kids offered enough concentrate diet before weaning [61]. In the current study, the inflection point for rumen weight was at 601.10 g and 157.46 d using the Gompertz function, suggesting that the optimum growth of rumen could be achieved after five months of age.

In this study, the derived AUC provided new insights about the growth over time. The AUC of pH was high till 28 d of age, then the pH decreased and the lowest pH was observed at 56 d. The decrease in pH could be due to solid feed supplementations after 28 d of age. The pH is important as an indicator of the ruminal development, the fermentation processes’ activity, and gut health. There are many factors that can influence the pH of the ruminal fluid, including feed intake, type of feed, and production state [64,65,66]. The transition from the liquid feed to the solid feed could be a reason of the pH reduction [67]. In the current study, the depression of pH was mitigated after 56 d. Several studies showed that ruminal development creates a state of adaptation of the ruminal mucosa and stabilizes pH [64,68].

The current study suggested that the growth curves could be used for better visualizing growth stages in the digestive tract over time. The growth stage is the repeated effect in the model for each individual [69]. the rumen, omasum, and reticulum percentages relative to the whole stomach showed increasing response with age (Figure 4). However, the percentage of abomasum declined with age. In another study, the rumen weight as a percentage of body weight was higher in 70 and 90 d kids than the young kids [70]. Similar results were observed in lambs [71] and calves [72,73]. The growth hormone receptor mRNA expression was higher in the rumen than in the other sections, especially the abomasum section in 42 d old lambs. Consequently, the enlargement of the abomasum was lower than the other parts [74]. The abomasum weight was higher during the monogastric phase in calves consuming only milk than those consuming solid feed plus milk because of the abomasum is the only compartment for milk digestion [64]. However, calves consuming only milk had less rumen development than those fed milk plus solid feed which showed significant increase in rumen volume and weight [75].

According to the Pearson correlation, the digestive tract development was more correlated with age than body weight. Hence, age had significant role than the animal weight for the growth of the digestive tract. The age effects are concerned with the differential of tissue development during the early stage of life, the physiological status of the animal, and its digestive capabilities [76]. Jiao et al. [8] examined the effects of age and feeding system in rumen weight in goats from birth to 70 d. They noted that the rumen weight increased via a temporal and successional process with age (*p* < 0.01) irrespective of the feeding system (*p* > 0.10).

## 5. Conclusions

The study showed the possibility of using the sigmoidal and polynomial models for describing the body growth and digestive tract development in goat kids. It is noteworthy that the MMF model provided the best overall goodness of fit for the development rumen and abomasum percentages over time. The model estimated parameters could be used for the prediction of the growth characteristics. Knowledge of the rumen development characteristics could help define the optimal management system, and feed regimes. However, the estimated parameters could be affected by the model types, the genotype of animals, and the environmental changes. Therefore, modeling the growth curves for different breeds in different environments will dissolve the problems we would face to obtain more accurate results.

## Figures and Tables

**Figure 1 animals-11-00757-f001:**
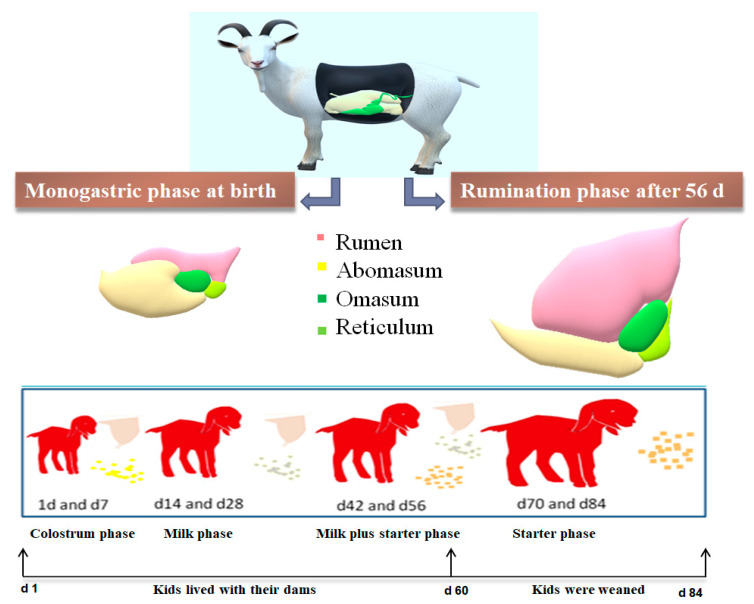
Feed management of goat kids from birth to postweaning according to the stages of rumen development.

**Figure 2 animals-11-00757-f002:**
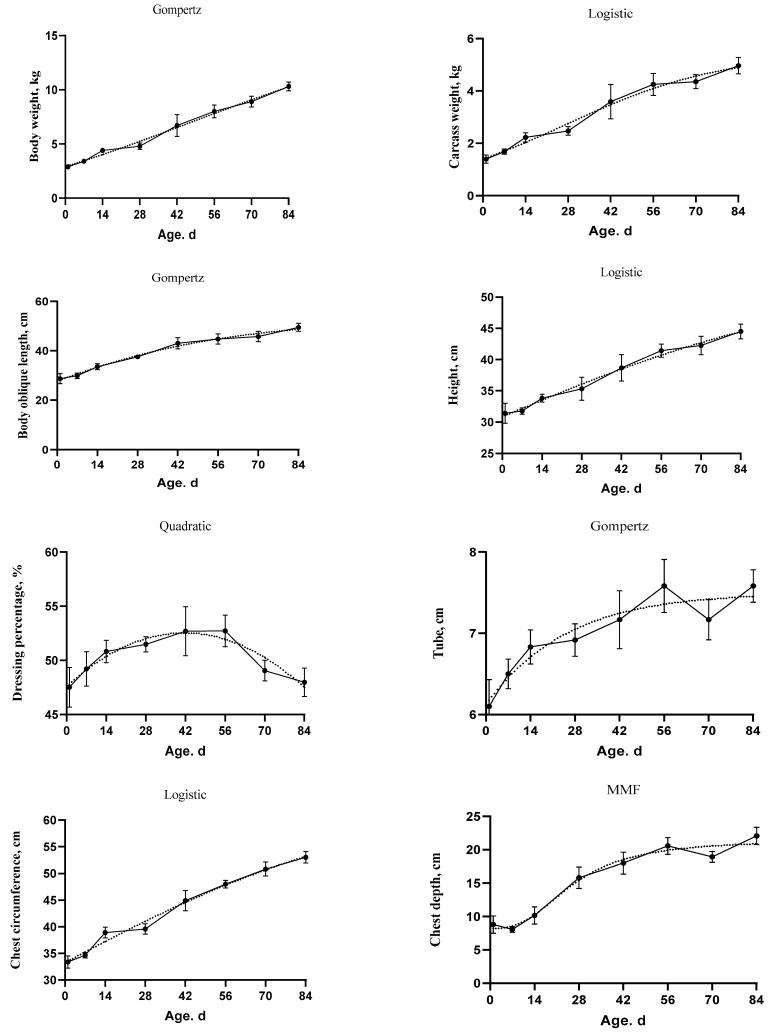
The fitted growth curves (dotted lines) of body weight, carcass weight, dressing percentage, and body size indices in goat kids based on the best models. The black points at the solid line with error bars represent the mean values with SEM of the observed data (*n* = 48).

**Figure 3 animals-11-00757-f003:**
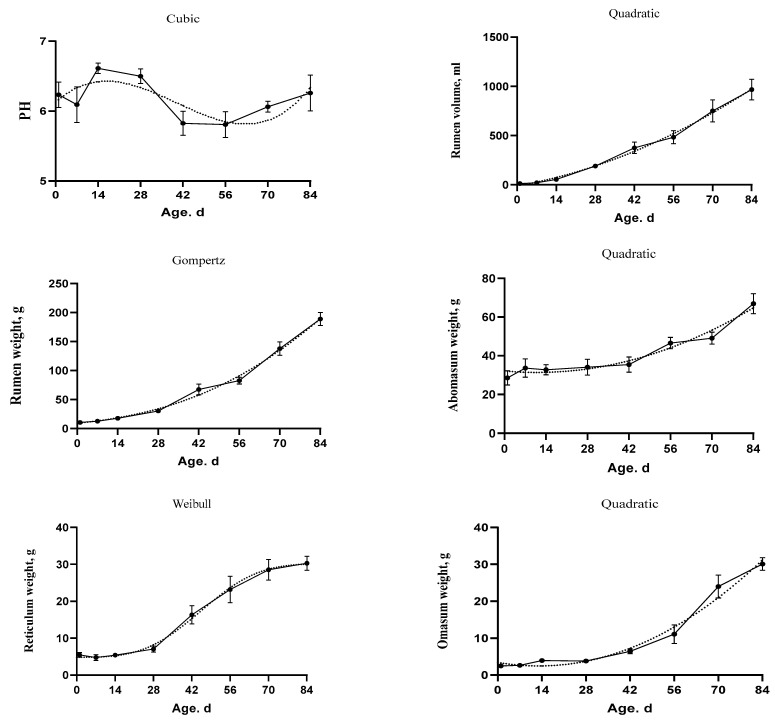
The fitted growth curves of the stomach development and rumen pH in goat kids based on the best models. The black points at the solid line with error bars represent the mean values with SEM of the observed data (*n* = 48).

**Figure 4 animals-11-00757-f004:**
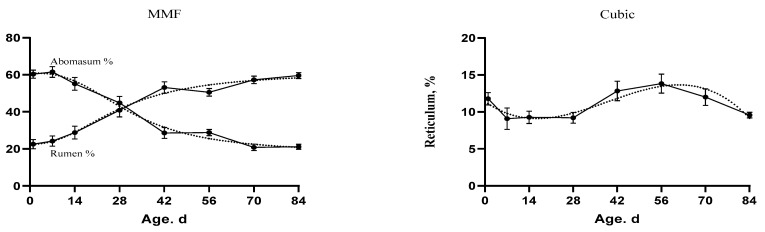
The fitted growth curves of the stomach compartments as percentage of the whole stomach weight in goat kids based on the best models. The black points at the solid line with error bars represent the mean values with SEM of the observed data (*n* = 48).

**Table 1 animals-11-00757-t001:** Ingredients and chemical analysis of the experimental solid diet (as a percentage of of dry matter).

Chemical Composition	%	Ingredients	%
Dry matter	95.25	Corn	59.8
Total energy, (MJ/kg)	18.03	Bran	11
Crude protein	19.63	Soybean meal	23
Crude fat	3.40	NaHCO_3_	0.5
Neutral detergent fiber	29.62	Premix	5
Acid detergent fiber	8.64	MgCl	0.7
Non-fibrous carbohydrate	39.36		
Ash	7.99		
Calcium	0.95		
Phosphorus	0.70		

**Table 2 animals-11-00757-t002:** Sigmoidal and polynomial models used in this study for modeling the growth curve of goat kids.

**Name of Model**	**Family of Model**	**Parameters Numbers**	Equation ^1^	Ipy ^2^	IPx
Logistic	Sigmoidal	3	$y=a/\left(1+{\mathrm{be}}^{-\mathrm{cx}}\right)$	a/2	(lnb)/c
Gompertz	Sigmoidal	3	y=a×exp(−exp(b−c×x))	a/e	b/c
MMF	Sigmoidal	4	$y=\frac{\mathrm{ab}+{\mathrm{cx}}^{d}}{b+x^{d}}$	(d(c + a) + (a − c))/2d	(b(d − 1)/d + 1)^1/d
Weibull	Sigmoidal	4	$y=a-{\mathrm{be}}^{-{\mathrm{cx}}^{d}}$	a − b × exp(1 − d/d)	(d − 1/cd)^1/d
Richards	Sigmoidal	4	$y=a/{\left(1+e^{b-\mathrm{cx}}\right)}^{1/d}$	a(1 + d)^ − 1/d	(b − ln d)/c
Ratkowsky	Sigmoidal	3	$y=a/\left(1+e^{b-\mathrm{cx}}\right)$	-	-
Quadratic	Polynomial	3	$y=a+\mathrm{bx}+{\mathrm{cx}}^{2}$	-	-
Cubic	Polynomial	4	$y=a+\mathrm{bx}+{\mathrm{cx}}^{2}+{\mathrm{dx}}^{3}$	-	-

^1^ y is the dependent variable; x refers to the independent variable. The parameters a, b, c, … n were used to define the scale and shape of the model curve, a is the asymptotic maximum value, b characterizes scaling parameter (constant of integration), c describes the instantaneous growth rate, d describes the shape parameter determining the position of the curve point inflection. For the Polynomials functions, a coefficient describing the intercepts, b to n refer to the intraspecific regression coefficients. ^2^ IPy: inflection point of y, IPx: inflection point of x.

**Table 3 animals-11-00757-t003:** The best goodness of fit of growth curve models for body weight before slaughter (kg), carcass weight (kg), dressing percentage (% of body weight), and body size indices (cm) of goat kids from birth to postweaning (*n* = 48).

**Items**	**Factors ^1^**	Gompertz	Logistic	Ratkowsky	MMF	Richards	Weibull	Quadratic	Cubic
Body Weight, kg	SEM	1.22	1.22	1.22	1.23	1.23	1.23	1.22	1.23
	AICc	19.88	19.89	19.89	22.21	22.16	22.21	19.94	22.17
	AIC	19.60	19.61	19.61	21.66	21.60	21.66	19.67	21.61
	BIC	23.30	23.31	23.31	27.21	27.15	27.21	23.37	27.16
Carcass, kg	SEM	0.79	0.79	0.79	0.80	0.80	0.80	0.79	0.80
	AICc	−21.31	−21.41	−21.41	−18.95	−19.14	−19.00	−21.16	−19.02
	AIC	−21.58	−21.68	−21.68	−19.50	−19.70	−19.56	−21.43	−19.58
	BIC	−17.88	−17.98	−17.98	−13.95	−14.15	−14.01	−17.73	−14.03
Dressing percentage, %	SEM	3.70	3.70	3.70	3.74	3.74	3.74	3.39	3.43
	AICc	124.21	124.21	124.21	126.34	126.33	126.33	115.97	118.12
	AIC	123.94	123.93	123.93	125.78	125.78	125.78	115.70	117.56
	BIC	127.64	127.63	127.63	131.33	131.33	131.33	119.40	123.11
Oblique length, cm	SEM	4.00	4.00	4.00	4.03	4.04	4.04	4.00	4.04
	AICc	131.38	131.40	131.40	133.45	133.67	133.58	131.56	133.60
	AIC	131.11	131.13	131.13	132.89	133.11	133.03	131.29	133.05
	BIC	134.81	134.83	134.83	138.44	138.66	138.58	134.99	138.60
Height, cm	SEM	3.24	3.24	3.24	3.28	3.28	3.28	3.25	3.28
	AICc	111.81	111.79	111.79	113.91	113.96	113.90	111.82	113.95
	AIC	111.54	111.51	111.51	113.35	113.40	113.34	111.55	113.39
	BIC	115.24	115.21	115.21	118.90	118.95	118.89	115.25	118.94
Tube, cm	SEM	0.62	0.62	0.62	0.62	0.63	0.62	0.62	0.63
	AICc	−43.80	−43.77	−43.77	−41.86	−40.85	−41.86	−43.29	−41.70
	AIC	−44.07	−44.04	−44.04	−42.41	−41.41	−42.41	−43.57	−42.25
	BIC	−40.37	−40.34	−40.34	−36.86	−35.86	−36.86	−39.87	−36.70
Circumference of chest, cm	SEM	2.78	2.78	2.78	2.81	2.81	2.81	2.78	2.81
	AICc	97.31	97.29	97.29	99.63	99.55	99.62	97.31	99.53
	AIC	97.04	97.02	97.02	99.08	98.99	99.06	97.03	98.97
	BIC	100.74	100.72	100.72	104.63	104.54	104.61	100.73	104.52
Chest depth, cm	SEM	3.11	3.09	3.09	3.05	3.11	3.08	3.13	3.17
	AICc	107.85	107.20	107.20	107.24	109.06	108.10	108.51	110.79
	AIC	107.58	106.93	106.93	106.68	108.51	107.54	108.24	110.23
	BIC	111.28	110.63	110.63	112.23	114.06	113.09	111.94	115.78

^1^ SEM: the mean square error; AICc: the corrected Akaike’s information criteria; AIC: the Akaike’s information criteria; BIC: Bayesian information criterion.

**Table 4 animals-11-00757-t004:** Estimated parameters (standard error), inflection points of y and x, and area under curve (AUC) for the best models related to the growth of weight and body measurements in goat kids (*n* = 48).

Variables ^1^	Model Name	Model Parameters ^2^				R^2^	IPy	IPx		AUC ^3^	
		a	b	c	d				1:28 d	29:56 d	57:84 d
Body weight, kg	Gompertz	18.61 (9.21)	0.62 (0.23)	0.01 (0.01)	-	0.82	6.85	45.93	110.21	177.25	245.79
Carcass, kg	Logistic	5.44 (0.74)	2.89 (0.55)	3.9 × 10^−2^ (0.01)	-	0.73	2.72	27.35	55.88	93.99	123.06
Dressing, %	Quadratic	47.61 (1.11)	0.24 (0.07)	−2.82 × 10^−3^ (7.8 × 10^−4^)	-	0.23	-	-	1357.04	1413.47	1350.42
Oblique length, cm	Gompertz	53.27 (4.86)	−0.43 (0.12)	2.37 × 10^−2^ (8.4 × 10^−3^)	-	0.77	19.60	−18.14	901.79	1130.41	1270.52
Height, cm	Logistic	54.57 (14.78)	0.77 (0.45)	1.46 × 10^−2^ (9.5 × 10^−3^)	-	0.69	27.29	−17.90	906.40	1040.33	1153.82
Tube, cm	Gompertz	7.62 (0.25)	−1.69 (0.21)	2.84 × 10^−2^ (0.03)	-	0.32	2.80	−59.51	180.53	195.42	200.23
Circumference of chest, cm	Logistic	64.27 (8.94)	0.93 (0.24)	1.78 × 10^−2^ (5.7 × 10^−3^)	-	0.86	32.14	−4.08	1008.37	1205.88	1369.31
Chest depth, cm	MMF	8.21 (1.09)	6.3 × 10^3^ (2.4 × 10^4^)	21.45 (1.60)	2.68 (1.15)	0.75	12.36	19.56	295.64	495.13	554.19

^1^ The model was built to fit the variables to the age of kids. ^2^ a, b, …, n: parameters that defined the scale and shape of the model curve, a: asymptotic maximum value, b: characterizes scaling parameter (constant of integration), c: maturing index, d: the shape parameter determining the position of the curve point inflection, R^2^: Coefficient of determination, IPy: inflection point of the dependent variable, IPx: age at the inflection point. ^3^ AUC: area under curve.

**Table 5 animals-11-00757-t005:** Pearson correlation coefficient (PCC) between body growth measurements and age or body weight (BW) of goat kids from birth to postweaning (*n* = 48).

Items		PCC ^1^	
	Age		BW
Body weight, kg	0.90 **		1.00
Carcass, kg	0.85 **		0.98 **
Dressing, %	0.00		0.19
Oblique length, cm	0.87 **		0.89 **
Height, cm	0.83 **		0.90 **
Tube, cm	0.56 **		0.75 **
Circumference of chest, cm	0.93 **		0.97 **
Chest depth, cm	0.83 **		0.84 **

^1^ Pearson’s coefficients with superscripts refer to the probability levels for significance tests (** *p* < 0.01), but those values without superscripts are not significant.

**Table 6 animals-11-00757-t006:** The best goodness of fit of growth curve models for ruminal fluid pH, rumen volume, stomach parts percentages of goat kids during the period from birth to postweaning (*n* = 48).

**Items**	Factors ^1^	Gompertz	Logistic	Ratkowsky	MMF	Richards	Weibull	Quadratic	Cubic
pH	SEM	0.56	0.55	0.55	0.56	0.56	0.55	0.55	0.52
	AICc	−53.12	−54.79	−54.54	−52.66	−52.26	−53.02	−55.49	−58.52
	AIC	−53.39	−55.06	−54.81	−53.22	−52.82	−53.58	−55.76	−59.08
	BIC	−49.69	−51.36	−51.11	−47.67	−47.27	−48.03	−52.06	−53.53
Rumen volume (mL)	SEM	149.36	151.11	151.11	159.54	379.43	150.24	148.67	150.33
	AICc	471.77	472.86	472.86	479.17	560.61	473.53	471.34	473.58
	AIC	471.50	472.59	472.59	478.61	560.05	472.97	471.06	473.02
	BIC	475.20	476.29	476.29	484.16	565.61	478.52	474.76	478.58
Rumen (g)	SEM	17.35	18.26	17.41	23.22	66.27	17.59	17.37	17.55
	AICc	269.43	274.21	269.76	298.00	396.59	271.90	269.52	271.67
	AIC	269.16	273.94	269.49	297.45	396.03	271.34	269.25	271.11
	BIC	272.86	277.64	273.19	303.00	401.58	276.89	272.95	276.66
Reticulum (g)	SEM	5.01	4.89	4.89	5.21	4.86	4.76	5.17	4.77
	AICc	152.69	150.37	150.37	157.54	150.91	149.00	155.58	149.18
	AIC	152.42	150.09	150.09	156.98	150.35	148.44	155.30	148.62
	BIC	156.12	153.80	153.80	162.53	155.90	153.99	159.01	154.17
Omasum (g)	SEM	4.09	4.13	4.03	5.55	3.99	4.30	3.99	4.04
	AICc	133.54	134.38	132.12	163.46	132.41	139.42	131.26	133.55
	AIC	133.27	134.11	131.85	162.90	131.85	138.87	130.99	132.99
	BIC	136.97	137.81	135.55	168.45	137.40	144.42	134.69	138.54
Abomasum (g)	SEM	9.71	9.64	9.64	10.35	9.75	9.26	9.22	9.22
	AICc	214.80	214.16	214.17	222.04	216.44	211.63	209.97	211.14
	AIC	214.53	213.89	213.90	221.48	215.89	211.07	209.69	210.58
	BIC	218.23	217.59	217.60	227.03	221.44	216.62	213.39	216.13
Rumen (%)	SEM	6.71	6.67	6.67	6.64	6.73	6.69	6.75	6.83
	AICc	180.05	179.50	179.50	180.34	181.63	181.10	180.66	182.94
	AIC	179.78	179.23	179.23	179.79	181.08	180.54	180.39	182.38
	BIC	183.48	182.93	182.93	185.34	186.63	186.09	184.09	187.93
Reticulum (%)	SEM	2.98	2.97	2.97	3.00	2.96	3.07	2.90	2.57
	AICc	103.68	103.63	103.63	105.61	104.30	107.79	101.35	91.09
	AIC	103.41	103.36	103.36	105.06	103.74	107.24	101.07	90.53
	BIC	107.11	107.06	107.06	110.61	109.29	112.79	104.77	96.08
Omasum (%)	SEM	2.09	2.08	2.08	2.17	2.11	1.96	2.01	2.03
	AICc	70.44	70.13	70.14	75.40	72.42	65.69	66.74	68.89
	AIC	70.17	69.86	69.87	74.84	71.86	65.14	66.46	68.33
	BIC	73.87	73.56	73.57	80.39	77.41	70.69	70.16	73.88
Abomasum (%)	SEM	17.73	7.31	6.85	6.43	6.69	6.48	6.72	6.63
	AICc	271.43	188.12	182.01	177.32	181.03	177.98	180.29	180.26
	AIC	271.16	187.85	181.74	176.76	180.47	177.42	180.02	179.70
	BIC	274.86	191.55	185.44	182.31	186.02	182.97	183.72	185.25

^1^ The SEM: the mean square error; AICc: the corrected Akaike’s information criteria; AIC: the Akaike’s information criteria; BIC: Bayesian information criterion.

**Table 7 animals-11-00757-t007:** Estimated parameters (standard error), inflection points of y and x, and area under the curve (AUC) for the best models related to the digestive tract of goat kids (*n* = 48).

**Variables ^1^**	**Model Name**	Model Parameters ^2^				R^2^	IPy	IPx		AUC ^3^	
		a	b	c	d				1:28 d	29:56 d	57:84 d
pH	Cubic	6.12 (0.21)	0.04 (0.03)	−1.5 × 10^−3^ (7.3 × 10^−4^)	1.3 × 10^−5^ (5.6 × 10^−6^)	0.15	-	-	171.88	163.97	160.54
Rumen volume, mL	Quadratic	−11.23 (48.57)	5.02 (2.96)	0.08 (0.03)	-	0.84	-	-	2244.05	9470.16	20,068.87
Rumen, g	Gompertz	1.6 × 10^3^ (2.2 × 10^3^)	1.65 (0.20)	0.01 (5.0 × 10^−3^)	-	0.93	601.10	157.46	537.32	1625.15	3730.92
Reticulum, g	Weibull	30.22 (2.10)	25.36 (2.58)	2.8 × 10^−6^ (9.5 × 10^−6^)	3.25 (0.89)	0.83	17.54	45.61	154.01	426.13	761.47
Omasum, g	Quadratic	3.44 (1.30)	−0.15 (0.08)	0.01 (9.2 × 10^−4^)	-	0.87	-	-	76.57	213.81	584.40
Abomasum, g	Quadratic	32.14 (3.01)	−0.14 (0.18)	6.3 × 10^−3^ (2.1 × 10^−3^)	-	0.63	-	-	859.76	1025.79	1456.99
Rumen, %	MMF	22.52 (2.53)	1.9 × 10^3^ (4.9 × 10^3^)	61.55 (5.16)	2.26 (0.80)	0.83	33.38	18.73	818.85	1340.07	1536.09
Reticulum, %	Cubic	11.41 (1.05)	−0.31 (0.13)	0.01 (3.6 × 10^−3^)	−1.0 × 10^-4^ (2.8 × 10^−5^)	0.31	-	-	258.94	319.86	336.97
Omasum, %	Weibull	10.06 (1.12)	4.69 (1.25)	3.7 × 10^-10^ (5.1 × 10^−9^)	5. 21 (3.40)	0.46	7.97	61.78	145.53	162.88	241.43
Abomasum, %	MMF	60.93 (2.15)	1.1 × 10^4^ (3.2 × 10^4^)	17.99 (4.63)	2.70 (0.85)	0.87	47.42	23.78	1477.44	869. 90	611.65

^1^ The model was built to fit the variables to the age of kids. ^2^ a, b, …, n: parameters that defined the scale and shape of the model curve, a: asymptotic maximum value, b: characterizes scaling parameter (constant of integration), c: maturing index, d: the shape parameter determining the position of the curve point inflection, R^2^: Coefficient of determination, IPy: inflection point of the dependent variable, IPx: age at the inflection point. ^3^ AUC: area under the curve.

**Table 8 animals-11-00757-t008:** Pearson correlation coefficient (PCC) between the measurements of digestive tract growth and age or body weight (BW) of goat kids from birth to postweaning (*n* = 48).

Items		PCC ^1^	
	Age		BW
Rumen pH	−0.17		−0.12
Rumen volume, mL	0.91 **		0.87 **
Rumen, g	0.94 **		0.90 **
Reticulum, g	0.89 **		0.85 **
Omasum, g	0.87 **		0.81 **
Abomasum, g	0.74 **		0.73 **
Rumen, %	0.88 **		0.80 **
Reticulum, %	0.20		0.15
Omasum, %	0.60 **		0.52 **
Abomasum, %	−0.90 **		−0.81 **

^1^ Pearson’s coefficients with superscripts refer to the probability levels for significance tests (** *p* < 0.01), but those values without superscripts are not significant.

## Data Availability

All data presented in this study are available on request from the corresponding author.

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
