# Peer review of "Predicting the Digestive Tract Development and Growth Performance of Goat Kids Using Sigmoidal Models"

_animals, 2021, doi:10.3390/ani11030757_

Round 1
Reviewer 1 Report
L146-L148: Please denote that these calculation of AIC, AICc, and BIC assumed that the errors followed iid normal distribution.
L160-L161: The authors must explain why AUC was calculated in this manuscript.
It might be better to calculated the derivatives of the best-fit models.
Table 2: Please add the number of parameters for Quadratic (4) and Cubic models (5).
English has to be checked entire the manuscript.
Author Response
Response to Reviewer 1 Comments
Point 1: L146-L148: Please denote that these calculation of AIC, AICc, and BIC assumed that the errors followed iid normal distribution.
Response 1: L156-157:It was assumed that the calculation of AIC, AICc, and BIC followed IID normal distribution (Independent and Identically Distributed Data).
Point 2: L160-L161: The authors must explain why AUC was calculated in this manuscript. It might be better to calculated the derivatives of the best-fit models.
Response 2: L165-168: In addition, the model could predict the area under the curve (AUC) between different critical points. AUC could describe the variation of growth as a function of time. AUC as a biological value is an important derivative of the models which was calculated as follows:
Point 3: Table 2: Please add the number of parameters for Quadratic (4) and Cubic models (5). Response 3: We updated it. Red colour
Point 4: English has to be checked entire the manuscript.
Response 4: We modified the language in the manuscript. Red colour.
Reviewer 2 Report
The authors revise the manuscript and answer my questions and change the focus of the paper to the mathematical model.
My original assessment was that the paper lack of necessary novelty and importance for publication in the journal and has limited scientific contribution to the field.
Changing the focus to mathematical models may soften these issues, especially considering the study in goats. Another point was regarding the sampling size, which may be limited (6 goats per group).
Since I am not an expert in mathematical models, I will leave the decision to approve or reject this manuscript to the editor taking into consideration the other reviewer's comments. I think that an opinion from an expert in mathematical models would be important here.
Minor comments:
L106: breast milk
Breast milk is woman milk, please revise.
Figure 1: please revise the text in the figure, some colors are difficult to read.
Author Response
Response to Reviewer 2 Comments
Point 1: L106: breast milk. Breast milk is woman milk, please revise. We updated it.
Point 2: Figure 1: please revise the text in the figure, some colors are difficult to read. We updated it in figure 1.
Point 3: The materials, methods and conclusion need to be improved. These parts were improved in the manuscript both language, style new references.

Reviewer 3 Report
Congratulations on the work, below I ask a series of questions in order to improve the work:
General comments:
Do you consider that six animals for each point is representative enough to be able to generalize?
Why is not reference to the settings of the models in the different figures? I think that is the most important when comparing them.
Why are there phrases in red? I imagine it is the corrections that you have made at a certain point, but please put them in black.
Normally more decimals are used when the numbers have fewer units, you always use the same decimal numbers, please unify criteria.
Sometimes the statistical error is shown in parentheses, sometimes it does not appear directly, please unify criteria.
Tables and especially graphs must be self-explanatory. All the information, the units and the number of data must appear, in addition to explaining what we are seeing. medium with standard error, confidence interval etc.
We are talking about evolution of the digestive tract, and this is obviously closely related to nutrition, however practically nothing is said about how this can affect it, I think that going deeper this could enrich the introduction and discussion.
Specific comments:
L54. Please remove the extra punctuation mark
L60. “The gastrointestinal tract grew from 13 uo to…” Please rewrite this phrase. It is not intuitive and I do not know what it has to do with the introduction
L70. Is? fits ...
L103. no more information on wellness?
L106. “the kids lived…” this information is not showed in Figure 1.
L106. according to the image at first they took colostrum
L110. they are separated from their mothers on day 60, but this does not appear in the image.
L111. In the image it says that from day 56 they eat only I think, however in the text they assure that they remain drinking milk.
L117. Ad libitum italics
L119. It does not seem intuitive to me, it is not an experimental design, they mix the development of the rumen, stomach etc ... if it is an experimental design, more data should appear, and have more consequence with what was developed in the previous paragraph.
L121. Please remove the bold "and"
L122. Lignine?
L124. How many animals?
L165. check the format, also where is the 1?
L197. The table appears on two pages, please put yourself on the same page.
L284. The figures look the same.
L418. I imagine that this "may" refers to the fact that they were adjusted correctly, so I think the logical thing is to explain the adjustments better. In Figures.
Author Response
Response to Reviewer 3 Comments
Dear reviewer:
Thank you very much for your careful review and constructive suggestions with regard to our manuscript. Those comments are helpful for authors to revise and improve our paper. We have studied comments carefully and tried our best to revise and improve the manuscript and made great changes in the manuscript according to the referees’ good comments. Revised portion is marked in red in the paper.
General comments:
Point 1: Do you consider that six animals for each point is representative enough to be able to generalize?
Response 1: Thanks for this important question. We think that six sheep for each age point were enough as they were selected from a large number of goats and slaughtered to represent the growth of each stage.
Point 2: Why is not reference to the settings of the models in the different figures? I think that is the most important when comparing them.
Response 2: Thanks very much for this important question. In this paper we firstly compared between 6 sigmoidal models included three-parameter models (Logistic, Gompertz, and Ratkowsky) and four-parameter models (MMF, Weibull, and Richards), as well as two polynomial models (Quadratic and cubic). We used SEM, AIC, AICc, and BIC for the evaluation of the models. Then, we selected only the best-fit model to be presented in our manuscript. The estimated model parameters, R2, IPy, IPx, and AUC were calculated and presented in tables. In addition, the compartment between the predicted data from the best model and the observed data were shown in figures. We think that eight models in one figure will be difficult to read and understand. In addition, the properties of other models were shown in tables. We attached two examples.
Point 3: Why are there phrases in red? I imagine it is the corrections that you have made at a certain point, but please put them in black.
Response 3: This is only way I have to prove that I did some modifications. I have some new parts red again due to the English correction and your comments.
Point 4: Normally more decimals are used when the numbers have fewer units, you always use the same decimal numbers, please unify criteria.
Response 4: I updated in figures (Tube and pH) and in tables about AUC
Point 5: Sometimes the statistical error is shown in parentheses, sometimes it does not appear directly, please unify criteria.
Response 5: I updated it.
Point 6: Tables and especially graphs must be self-explanatory. All the information, the units and the number of data must appear, in addition to explaining what we are seeing. medium with standard error, confidence interval etc.
Response 6: The title of graphs changed for example:
Figure 3. The fitted growth curves of the stomach development and rumen pH in goat kids based on the best models. The black points at the solid line with error bars represent the mean values with SEM of the observed data (n = 48).
Point 7: We are talking about evolution of the digestive tract, and this is obviously closely related to nutrition, however practically nothing is said about how this can affect it, I think that going deeper this could enrich the introduction and discussion.
Response 7: We updated it in the introduction and discussion
In introduction : After birth, the rumen has undeveloped volume and function, and the esophageal groove allows milk to enter the abomasum and keeps rumen out of milk [12]. The milk intake gradually decreased and the solid intake increased due to progressive independent behavior from the dam over time [13]. Solid feed had significant impact on the rumen development via increasing the rumen volatile fatty acids [8,14]
In discussion : The abomasum weight was higher during the monogastric phase in calves consuming only milk than those consuming solid feed plus milk because of the abomasum is the only compartment for milk digestion [64]. However, calves consuming only milk had less rumen development than those fed milk plus solid feed which showed significant increase in rumen volume and weight [75].
However, we also said about nutrition in ANOTHER PARTS in the manuscript.
Specific comments:
Point 8: L54. Please remove the extra punctuation mark.
Response 8: We updated it.
Point 9: L60. “The gastrointestinal tract grew from 13 uo to…” Please rewrite this phrase. It is not intuitive and I do not know what it has to do with the introduction.
Response 9: This phrase had been addressed to confirm that the gastrointestinal tract is an important organ as its high percentage and growth and the rumen is the most important part
In goats, the gastrointestinal tract as a percentage of the empty body weight increased from 13 up to 27% as they grew, in which the rumen contributed most to the growth [6]
Point 10: L70. Is? fits ...
Response 10: The livestock growth curve has a sigmoid shape so that it can be fitted and analyzed by the nonlinear mathematical model.
Point 11: L103. no more information on wellness?
Response 11: Sorry, I moved it to the part of * Institutional Review Board Statement:*
Experimental work was performed in accordance to the guidance approved by the Animal Ethics Committee of the Chinese Academy of Agricultural Sciences (Protocol number: AEC-CAAS-20191105; Approval date: 3 November 2019).
Point 12: L106. “the kids lived…” this information is not showed in Figure 1.
Response 12: We updated it in the figure 1.
Point 13: L106. according to the image at first they took colostrum.
Response 13: we updated it in the text.
As shown in Figure 1, the kids lived with dams and consumed colostrum from birth to d 7 then milk from d 8 to d 28. After 28 d of age, the concentrate was introduced to the kids as their supplementary diet. The kids were weaned at 60 d of age and separated from their dams.
Point 14: L110. they are separated from their mothers on day 60, but this does not appear in the image. We updated it in the figure.
Response 14: We updated it in figure 1.
Point 15: L111. In the image it says that from day 56 they eat only I think, however in the text they assure that they remain drinking milk.
Response 15: Updated.
Point 16: L117. Ad libitum italics.
Response 16: Updated
Point 17: L119. It does not seem intuitive to me, it is not an experimental design, they mix the development of the rumen, stomach etc ... if it is an experimental design, more data should appear, and have more consequence with what was developed in the previous paragraph.
Response 17: Changed, Figure 1. Feed management of goat kids from birth to postweaning according to the stages of rumen development.
Point 18: L121. Please remove the bold "and".
Response 18: Removed
Point 19: L122. Lignine?
Response 19: We did not measure lignin in the diet.
Point 20: L124. How many animals?
Response 20: Updated
The selected forty-eight goat kids were weighted before feeding.
Point 21: L165. check the format, also where is the 1?
Response 21: Updated in table 2
Point 22: L197. The table appears on two pages, please put yourself on the same page.
Response 22: Updated
Point 23: L284. The figures look the same.
Response 23:
The figures is different, Figure 3 was about the weight of the digestive tract parts, while figure 4 was about the digestive tract percentages.
Point 24: L418. I imagine that this "may" refers to the fact that they were adjusted correctly, so I think the logical thing is to explain the adjustments better. In Figures.
Response 24: I guess the reviewer means that the sentence of "The sigmoidal and polynomial models can describe the growth curves of the body weight and digestive tract development." is not a very appropriate conclusion. I updated it.
The study showed the possibility of using the sigmoidal and polynomial models for describing the body growth and digestive tract development in goat kids.
In addition, the materials, methods, results and conclusion were improved

Round 2
Reviewer 1 Report
> Point 1: L146-L148: Please denote that these calculation of AIC, AICc, and BIC assumed that the errors followed iid normal distribution.
> Response 1: L156-157:It was assumed that the calculation of AIC, AICc, and BIC followed IID normal distribution (Independent and Identically Distributed Data).
It was not assumed that AIC, AICc, and BIC themselves follow IID normal distribution. Therefore, please denote the fact more precisely.
Author Response
Author's Reply to the Review Report (Reviewer 1)
Dear reviewer:
Thank you very much for your careful review and constructive suggestions with regard to our manuscript. These comments help me much to improve my manuscript.
Point 1:
> Point 1: L146-L148: Please denote that these calculation of AIC, AICc, and BIC assumed that the errors followed iid normal distribution.
> Response 1: L156-157:It was assumed that the calculation of AIC, AICc, and BIC followed IID normal distribution (Independent and Identically Distributed Data).
It was not assumed that AIC, AICc, and BIC themselves follow IID normal distribution. Therefore, please denote the fact more precisely.
Response 1:
Thanks very much for this point and sorry because I mentioned the calculation themselves not the model errors.
The actual nonlinear model can be written as yi = F(xi, θ) + ϵi where with θ is the parameters vector, and ϵi is the random error. The assumptions of regression are: The errors are normally distributed and independent. The residual is an ‘approximation’ for the error term in practical analyses. The residuals of the model should be normally distributed; otherwise the regression model will not explain all trends in the dataset. So, we should find the model, that shows residuals which are normally distributed. To obtain the model parameter estimates, we have to make assumptions about the distribution of your residuals and this assumption is that the residuals are normally distributed. However, there is some literature that suggests (unless there are marked outliers or the samples were few) we do not need to get worked up about the Normality assumption
eg Misspecifying the Shape of a Random Effects Distribution: Why Getting It Wrong May Not Matter, Statistical Science, 2011, Vol. 26, No. 3, 388–402
So, we modified the sentence in the manuscript:
Line 156-157: The calculation of AIC, AICc, and BIC assumed that the model errors were independent with normal distribution.
Our modification is upon the first reviewer comment and some literature
eg Pinho, S. Z. D., Carvalho, L. R. D., Mischan, M. M., & Passos, J. R. D. S. (2014). Critical points on growth curves in autoregressive and mixed models. Scientia Agricola, 71(1), 30-37.
https://www.sciencedirect.com/topics/medicine-and-dentistry/akaike-information-criterion

Reviewer 3 Report
L156. It was assumed that the calculation of AIC, AICc, and BIC 156 followed IID normal distribution (Independent and Identically Distributed Data). that cannot be assumed, it has to be verified. Otherwise you cannot use this analysis
L172. table continues with form errors
Author Response
Author's Reply to the Review Report (Reviewer 1)
Dear reviewer:
Thank you very much for your careful review and constructive suggestions with regard to our manuscript. These comments help me much to improve my manuscript.
Point 1:
> Point 1: L146-L148: Please denote that these calculation of AIC, AICc, and BIC assumed that the errors followed iid normal distribution.
> Response 1: L156-157:It was assumed that the calculation of AIC, AICc, and BIC followed IID normal distribution (Independent and Identically Distributed Data).
It was not assumed that AIC, AICc, and BIC themselves follow IID normal distribution. Therefore, please denote the fact more precisely.
Response 1:
Thanks very much for this point and sorry because I mentioned the calculation themselves not the model errors.
The actual nonlinear model can be written as yi = F(xi, θ) + ϵi where with θ is the parameters vector, and ϵi is the random error. The assumptions of regression are: The errors are normally distributed and independent. The residual is an ‘approximation’ for the error term in practical analyses. The residuals of the model should be normally distributed; otherwise the regression model will not explain all trends in the dataset. So, we should find the model, that shows residuals which are normally distributed. To obtain the model parameter estimates, we have to make assumptions about the distribution of your residuals and this assumption is that the residuals are normally distributed. However, there is some literature that suggests (unless there are marked outliers or the samples were few) we do not need to get worked up about the Normality assumption
eg Misspecifying the Shape of a Random Effects Distribution: Why Getting It Wrong May Not Matter, Statistical Science, 2011, Vol. 26, No. 3, 388–402
So, we modified this sentence in the manuscript:
Line 156-157: The calculation of AIC, AICc, and BIC assumed that the model errors were independent with normal distribution.
Our modification is upon the first reviewer comment and some literature
eg Pinho, S. Z. D., Carvalho, L. R. D., Mischan, M. M., & Passos, J. R. D. S. (2014). Critical points on growth curves in autoregressive and mixed models. Scientia Agricola, 71(1), 30-37.
https://www.sciencedirect.com/topics/medicine-and-dentistry/akaike-information-criterion

This manuscript is a resubmission of an earlier submission. The following is a list of the peer review reports and author responses from that submission.
Round 1
Reviewer 1 Report
The major comments/questions for this manuscript were shown below.
Overall, I require the authors to improve the Materials and Methods section more correctly and in more detail.
P3L96-104, Figure 1: The day points shown in the sentences and Figure 1 was different (e.g., d30 in sentences v.s. d28 in Figure1). Which is correct?
P4L110-111: The authors should explain the conditions of goat kids before measuring their records, such as information including the timing of recording and feeding managements.
P4L112-113: The authors should explain how to measure the carcass weight and growth indicators of organs in more detail.
P4L120-121: The authors should explain the conditions of measuring rumen pH, because the rumen pH might greatly change within a short-term.
P4L138-140: The model with the lowest AIC value was always the same as that with the highest R2 value in this study?
Table 2: The authors should show all models compared in this study including linear ones.
P4L147-150: Sorry, but I cannot completely understand what the authors did here. Please explain in more detail.
Table 8: Independent → Dependent; stomach → stomach
Reviewer 2 Report
The manuscript entitled Longitudinal investigation of anatomic development and growth characteristics in goat kids from birth to postweaning from Abdelsattar and others is well written and has a well-described and robust methodology, but, in my opinion, lack the necessary novelty for publication in the journal. What is the scientific contribution of the work? How it will advance the field? What was the hypothesis? The authors fail to perform a more dense study comparing some factors, such as feeding type, breeds or so. This would provide some comparison and more insights on the subject. They just evaluated one homogenous group of animals during several points according to age. Additionally, for this type of study, I think 6 animals per group is a limited sample size considering the animal variation.
Observing the conclusion of the work: “the age differentially affected the growth of stomach compartments. The rumen percentage to the complex stomach was increasing with age…. However, the abomasum percentage to the complex stomach was decreasing over time.
Such information is well known in the field. The study lack novelty and importance.
Similar papers in the same field usually test some hypothesis, as the examples below:
https://doi.org/10.1016/j.smallrumres.2004.08.009
https://doi.org/10.2527/jas.2011-4500
https://link.springer.com/article/10.1007/s11250-018-1666-8
Mino comments:
Line 415: The present study investigated that the body weight, body size indexes, organs, and stomach parts increased gradually in goat kids raised according to the common and natural conditions.
You don´t need to repeat your objective in the conclusion, be objective about your results.
Figure 5 has a poor design with all those underlines and there is a problem in figure X-axis.
Why there are missing values in table 6? If the model cannot be applied this must be informed.
In figures when you have several parameters that summed to 100% the best way to represent in a graph is trough bar chart with stacked columns.
At table 4 you don´t mention the data used in the model. No mention to the age of the animals whatsoever. This must be informed in the table(s). The model was built to fit the variables to the age of kids.
Reviewer 3 Report
Dear authors,
the paper entitled “Longitudinal investigation of anatomic development and growth characteristics in goat kids from birth to postweaning”. has some critical points that make it not suitable for publication.
Some results regarding differences between “age” groups are not characterized by novelty, moreover there is probably a problem with the statistical approach. Differently, it is interesting the modelling approach to growth curves.
Remarks:
Abstract
Lines 27-28. Not clear what authors want to say. Please revise the sentence.
Material and Methods
In this case it is important to report not only the chemical composition of the feed, but also the individual intake (considering that they are only 6 for each age group) and the composition (how much hay? Quality? Dimensions?). There are a lot of lacks considering that authors want to investigate the organs growth (and particularly the rumen).
The statistical approach is not correct considering that there is not consideration of repeated measures (you have 8 age groups).
Considering these aspects, it is difficult to give a adequate evaluation of the entire discussion and conclusion sections, because they probably come from an incorrect approach to data analysis.